# Hedgehog-GLI and Notch Pathways Sustain Chemoresistance and Invasiveness in Colorectal Cancer and Their Inhibition Restores Chemotherapy Efficacy

**DOI:** 10.3390/cancers15051471

**Published:** 2023-02-25

**Authors:** Anna Citarella, Giuseppina Catanzaro, Zein Mersini Besharat, Sofia Trocchianesi, Federica Barbagallo, Giorgio Gosti, Marco Leonetti, Annamaria Di Fiore, Lucia Coppola, Tanja Milena Autilio, Zaira Spinello, Alessandra Vacca, Enrico De Smaele, Mary Anna Venneri, Elisabetta Ferretti, Laura Masuelli, Agnese Po

**Affiliations:** 1Department of Experimental Medicine, Sapienza University of Rome, Viale Regina Elena 324, 00161 Rome, Italy; 2Department of Molecular Medicine, Sapienza University of Rome, Viale Regina Elena 291, 00161 Rome, Italy; 3Faculty of Medicine and Surgery, Kore University of Enna, Cittadella Universitaria, 94100 Enna, Italy; 4Soft and Living Matter Laboratory, Institute of Nanotechnology, Consiglio Nazionale Delle Ricerche, Piazzale Aldo Moro 5, 00185 Rome, Italy; 5Center for Life Nano- and Neuro-Science, Istituto Italiano di Tecnologia, Viale Regina Elena 291, 00161 Rome, Italy; 6D-TAILS srl, Viale Regina Elena 291, 00161 Rome, Italy

**Keywords:** colorectal cancer, signaling pathways, chemoresistance, epithelial-to-mesenchymal transition, organoids

## Abstract

**Simple Summary:**

Colorectal cancer is a leading cause of cancer-related deaths, mainly caused by resistance to therapy and metastatic spread, in turn sustained by the activation of mechanisms such as the epithelial-to-mesenchymal transition (EMT). We investigate here the role of the Hedgehog-GLI and NOTCH signaling pathways, already associated with poor prognosis in CRC, in the mechanism of chemoresistance and EMT, using monolayer and organoids from two models of common mutations in CRC: KRAS and BRAF. Our results show that treatment with the chemotherapeutic drug 5-fluorouracil activated both pathways in the investigated contexts. However, we observed a different behavior in the investigated models: in KRAS-mutated CRC, the inhibition of both the HH-GLI and NOTCH pathways is necessary to enhance chemosensitivity, while in BRAF-mutated CRC the inhibition of HH-GLI is sufficient to impair both signaling pathways and promote chemosensitivity.

**Abstract:**

Colorectal cancer (CRC) is a leading cause of cancer-related mortality and chemoresistance is a major medical issue. The epithelial-to-mesenchymal transition (EMT) is the primary step in the emergence of the invasive phenotype and the Hedgehog-GLI (HH-GLI) and NOTCH signaling pathways are associated with poor prognosis and EMT in CRC. CRC cell lines harboring KRAS or BRAF mutations, grown as monolayers and organoids, were treated with the chemotherapeutic agent 5-Fluorouracil (5-FU) alone or combined with HH-GLI and NOTCH pathway inhibitors GANT61 and DAPT, or arsenic trioxide (ATO) to inhibit both pathways. Treatment with 5-FU led to the activation of HH-GLI and NOTCH pathways in both models. In KRAS mutant CRC, HH-GLI and NOTCH signaling activation co-operate to enhance chemoresistance and cell motility, while in BRAF mutant CRC, the HH-GLI pathway drives the chemoresistant and motile phenotype. We then showed that 5-FU promotes the mesenchymal and thus invasive phenotype in KRAS and BRAF mutant organoids and that chemosensitivity could be restored by targeting the HH-GLI pathway in BRAF mutant CRC or both HH-GLI and NOTCH pathways in KRAS mutant CRC. We suggest that in KRAS-driven CRC, the FDA-approved ATO acts as a chemotherapeutic sensitizer, whereas GANT61 is a promising chemotherapeutic sensitizer in BRAF-driven CRC.

## 1. Introduction

Colorectal cancer (CRC) is the third most frequent cancer and the second cause of cancer-related death worldwide [1]. Mutations in KRAS and BRAF oncogenes represent the most common genetic drivers in CRC. Indeed, KRAS and BRAF mutations occur in 40% and 10% of CRC, respectively [2], and they are both associated with a poor outcome [3]. Even though they both belong to the MAPK pathway, KRAS and BRAF mutations are mutually exclusive in CRC and these two types of cancer are characterized by distinct clinical and molecular features. BRAF-mutant CRC often displays genome-wide hypermethylation, high microsatellite instability and mutation rates, while KRAS mutant CRC is associated with lower levels of microsatellite instability and gene methylation [2].

First-line and palliative treatments for metastatic CRC, bearing KRAS or BRAF mutations, include the cytotoxic chemotherapeutic agent 5-fluorouracil (5-FU) [1,4]; however, patients often present disease recurrence after 5-FU therapy [5]. Chemoresistance is conferred by a plethora of mechanisms, including the modulation of signaling pathways involved in the emergence of the cancer stem features and epithelial-to-mesenchymal transition (EMT) [6]. Other mechanisms for resistance to therapy include the inhibition of apoptosis driven by upregulation of autophagy [7], metabolic reprogramming [8], upregulation of molecules involved in drug efflux and drug metabolism and activation of alternative pathways [9].

Hedgehog-GLI (HH-GLI) and NOTCH signaling are pivotal developmental pathways involved in the regulation of multiple biological and pathological processes. The canonical HH-GLI pathway is activated upon the interaction between the extracellular ligands Shh, Ihh and Dhh and the receptor Patched (PTCH), which in turn derepresses Smoothened (Smo), thus activating the transcription factors GLI1, GLI2 and GLI3. Activated GLI translocate into the nucleus where they bind to DNA and activate the transcription of target genes [10]. In cancers, GLI1 can also be activated in a non-canonical way by the “oncogenic load” of the cancer cell [11]. NOTCH cascade is activated upon binding of ligands Jag1, Jag2, Dll1, Dll3 and Dll4 to NOTCH receptors (from 1 to 4). The binding leads to proteolytic cleavages of the NOTCH receptor, releasing the NOTCH intracellular domain (ID) into the cytoplasm. Then, NOTCH ID migrates into the nucleus where, in complex with CBF1 (also known as RPBJ), it activates its transcriptional program [12]. Downstream target genes include HES1, which is involved in EMT and transcriptionally regulates ATP-binding cassettes transporters (ABC transporters), involved in multidrug resistance [13]. Interestingly, deregulation of the NOTCH pathway was described in numerous cancerous and non-cancerous diseases, with its role being highly context-dependent [14].

Deregulated HH-GLI is involved in the development and maintenance of numerous cancers [10] and, together with NOTCH signaling, plays a crucial role in the maintenance of stem cells of the intestinal epithelia [15]. The crosstalk of HH-GLI and NOTCH signaling is fundamental for spinal cord patterning [16], and several previous reports highlighted how several molecules belonging to the NOTCH pathway regulate the key components of the HH-GLI pathway and vice versa, as reviewed Kumar et al. [17].

Both HH-GLI and NOTCH pathways were described as deregulated and associated with poor prognosis in CRC [18,19]. In this context, our previous work has described a chemoresistance mechanism operated by the HH-GLI signaling in CRC, where chemotherapy treatment resulted in aberrant activation of the HH-GLI pathway which in turn led to the transcription of ATP-binding cassette transporters (ABC transporters), involved in multidrug resistance [20].

Therefore, our current work aimed to evaluate the role of HH-GLI and NOTCH signaling pathways as regulatory molecular mechanisms responsible for chemotherapy resistance in models of KRAS- or BRAF-driven CRC.

## 2. Materials and Methods

### 2.1. Cell Cultures and Treatments

HCT116 (*KRAS G13D* mutant) and HT29 (*BRAF V600E* mutant) were obtained from American Type Culture Collection (ATCC) and grown in DMEM high glucose (supplemented with 10% (*v*/*v*) fetal bovine serum, 1% (*v*/*v*) penicillin (50 U mL^−1^)—streptomycin (50 U mL^−1^)—and 2 mM L-glutamine. Cells were routinely checked for mycoplasma contamination by testing with PCR Mycoplasma Detection Kit (Cat. G238, ABM, Richmond, BC, Canada).

Cells were treated with 10 μM GANT61 (ENZO Lifesciences, New York, NY, USA), 10 μM DAPT (Merk Life Science S.r.l., Milan, Italy), 10 μM Arsenic Trioxide (ATO) (Merck, Merk Life Science S.r.l., Milan, Italy) and 10 μM 5-Fluorouracil (5-FU).

For combined treatments, GANT61 and DAPT or Arsenic Trioxide (ATO) were administered to the cells 24 h before 5-FU.

### 2.2. Cell Viability by Trypan Blue Exclusion Assay

Cell proliferation was assessed by trypan blue dye exclusion test using 0.4% (*w*/*v*) Trypan Blue solution (Merk Life Science S.r.l., Milan, Italy). Blue-stained cells were scored as non-viable and unstained cells were scored as viable cells. The percentage of viable cells was obtained as the ratio between the percentage of viable cells in treated cells versus control.

### 2.3. Transwell Invasion Assay

Transwell invasion assay was performed using Corning^®^ Transwell^®^ chambers (8 μm pore size, Corning^®^). HCT116 and HT29 cells (2.5 × 10^4^ in each well) were seeded in the upper chambers of the 48-well plates (Corning, Somerville, MA, USA) while lower chambers were filled with 1 mL of medium with indicated treatments. Cells in the lower chambers were fixed with 95% ethanol for 10 min, stained with crystal violet and counted.

### 2.4. Western Blot

Cells were lysed as previously described [20]. Lysates were separated on 8% acrylamide gel and immunoblotted using standard procedures [21]. Primary antibodies were Anti-GLI1 (L42B10, Cell Signalling Technology Inc., Boston, MA, USA), anti-PARP p85 Fragment (G7341, Promega, Madison, WI, USA) and anti-Cleaved NOTCH1 (D3B8, Cell Signalling Technology Inc., Boston, MA, USA). HRP-conjugated secondary antisera (Santa Cruz Biotechnology, Shanghai, China) were used, followed by enhanced chemiluminescence (ECL Amersham, Merk Life Science S.r.l., Milan, Italy).

### 2.5. RNA Isolation and Real Time qPCR

cDNA was obtained as described earlier [20]. RNA expression was analyzed on cDNAs using the ViiA™ 7 Real-Time PCR System, SensiFAST™ Probe Lo-ROX (Bioline, Memphis, TN, USA), TaqMan gene expression assay according to the manufacturer’s instructions (Life Technologies, Waltham, MA, USA). mRNA quantification was expressed in arbitrary units, as the ratio of the sample quantity to the calibrator or to the mean values of control samples. All values were normalized to three endogenous controls: HPRT, GAPDH and β-ACTIN.

Primers for gene expression are listed in Appendix A. Gene expression of GLI1, HES1, c-MET, ABCG2, CD133, KRAS, BRAF, HPRT, GAPDH and β-ACTIN was assessed using Life technologies “best coverage” assays (Life Technologies, Waltham, MA, USA).

### 2.6. Organoids

Organoids were produced by seeding 1500 cells per well. Cells were mixed with 33% growth-factor-reduced phenol red-free Matrigel (Corning, Somerville, MA, USA). Cultures were grown using a flat-bottom 24-well microplate in advanced DMEM-F12 (Cat. 12634010, Gibco, Waltham, MA, USA) supplemented with Epidermal growth factor and Fibroblast growth factor both at final concentrations of 20 ng/µL.

For in vivo live imaging experiments, GFP-labelled organoids were obtained by transducing HCT116 with PLKO lentiviral particles carrying pTWEEN-GFP vector.

Transduced green fluorescent cells were selected by cell sorting and used for organoids production.

### 2.7. Whole Mount Immunofluorescence

Organoids were fixed with 4% paraformaldehyde and permeabilized with Triton X-100 in PBS (Sigma-Aldrich, St. Louis, MO, USA). Organoids were stained with anti-vimentin (ab11256, ABCAM, Cambridge, UK) antibody. Nuclei were DAPI-counterstained. Phalloidin was used for f-actin staining. Images were acquired using an LSM 900 (Zeiss, Milan, Italy) laser scanning confocal microscope with 40×/0.75 NA objective. Images were analyzed by using the program Zeiss ZEN 2.3 blue edition (https://www.zeiss.com/microscopy/int/products/microscope-software/zen-lite.html (accessed on 10 September 2022)).

### 2.8. Datasets and In Silico Analyses

Datasets available on R2 platform (https://hgserver1.amc.nl/cgi-bin/r2/main.cgi (accessed on 15 December 2021)) were interrogated to evaluate GLI1 and NOTCH1 correlation in patients carrying BRAF or KRAS mutations. In detail, Tumor Colon Mutation status (Core Exon)—Sieber—211—rma_sketch—huex10p investigated gene correlation between GENE/REPORTER1: GLI1 and GENE/REPORTER2: NOTCH1, in 29 samples of CRC-carrying braf_v600e mutation; Tumor Colon (after surgery)—Beissbarth—363—custom—4hm44k investigated gene correlation between GENE/REPORTER1: GLI1 and GENE/REPORTER2: NOTCH1, in 32 samples of CRC carrying kras_g13d mutation.

### 2.9. Single-Particle Tracking Analysis

Single-particle tracking (SPT) diffusibility analysis was performed in five steps. In the first step, single cells were detected from the time-lapse movies of oHCT116-treated and control group organoids using Imaris spot model. Spots were taken in each frame and were linked to the spots corresponding to the same cell in the successive frame. Frame-to-frame tracking was implemented using the linear assignment problem (LAP) method [22,23]. In the third step, MSD, the mean square distance travelled by a cell given a certain time interval (see Appendix A), was computed from single trajectories, as described by Michalet X [24]. In the fourth step, the diffusion parameter D was calculated for each tracked cell. To this end, the MSD was plotted for different time intervals (Δt) for each cell trajectory and the slope was computed using the Least Squares Method. In the fifth step, the Kolmogorov–Smirnov test was applied on the diffusion parameters D, obtained from 5-FU-treated oHCT116 and control group organoids.

### 2.10. Statistical Analysis

Results are representative of at least three independent experiments and are expressed as means +/− SD. Differences were analyzed using One-way ANOVA and Two-way ANOVA tests where appropriate, using the GraphPad Prism software Version 8.0. Adjusted *p*-values of less than 0.05 were considered as statistically significant.

## 3. Results

### 3.1. HH-GLI and NOTCH Signaling Pathways Sustain Resistance to 5-FU in KRAS Mutant CRC Cells

5-fluorouracil (5-FU) is a chemotherapeutic agent used for adjuvant and palliative treatment of CRC; however, patients often present disease recurrence [5]. Therefore, we evaluated the role of HH-GLI and NOTCH signaling pathways as molecular mechanisms responsible for chemotherapy resistance.

KRAS mutant HCT116 CRC cells were treated with 5-FU, alone or in combination with the HH-GLI inhibitor GANT61 and/or the NOTCH inhibitor DAPT. Our results showed that GLI1 and NOTCH1 ID were significantly upregulated after 5-FU treatment (Figure 1A). HH-GLI inhibition by GANT61 resulted in the downregulation of GLI1 and, interestingly, in the upregulation of NOTCH1 ID; vice versa, NOTCH inhibition by DAPT resulted in the downregulation of NOTCH1 ID and the upregulation of GLI1 (Figure 1A).

We then determined the effects of treatments with the combination of HH-GLI inhibitor, NOTCH inhibitor and 5-FU. NOTCH1 ID expression was impaired in all combined treatments that included DAPT (DAPT+GANT61, DAPT+5-FU and DAPT+GANT61+5FU), while it was unaffected by the combination of 5-FU+GANT61.

On the other hand, GLI1 was downregulated by GANT61 alone, as well as when combined with 5-FU and 5-FU+DAPT, while the combination of NOTCH inhibition and 5-FU failed to inhibit GLI1.

Overall, our results show that GLI1 and NOTCH1 ID levels were concomitantly significantly downregulated only after the combined treatment of the chemotherapeutic agent 5-FU together with the inhibition of HH-GLI and NOTCH.

In addition, we observed that the combination of HH-GLI and NOTCH pathway inhibition prevents the GLI1 upregulation and NOTCH1 activation induced by 5-FU.

To determine the effects of treatments on apoptosis, levels of cleaved PARP (c-PARP) were evaluated; our results show that c-PARP was significantly induced by the combination of 5-FU and GANT61 or DAPT, and the three drugs combined (Figure 1A).

We further analyzed the effects of treatments on cell viability, and we found it significantly impaired in cells treated with the combination of GANT61 and DAPT and with the combination of 5-FU with either GANT61 or DAPT or the combination of the three drugs (Figure 1B). 

To discern potential interdependence between the HH-GLI and NOTCH signaling pathways in mutant KRAS CRC cells, we analyzed GLI1 and NOTCH1 levels in an available cohort of CRC patients carrying this mutation (Tumor Colon (after surgery)—Beissbarth—363—custom—4hm44k; https://hgserver1.amc.nl/cgi-bin/r2/main.cgi, accessed on 15 December 2021) and no correlation was found (Figure 1C)

We therefore envisioned a model where the oncogenic force of the driver gene *KRASG13D* sustains both HH-GLI and NOTCH pathways and both pathways need to be targeted to achieve a successful impairment of cells after chemotherapy. 

Hence, to clarify if the KRASG13D driver mutation sustained expression of GLI1 and NOTCH, we performed silencing of KRAS in HCT116 (Appendix A), which resulted in the significant downregulation of GLI1 and NOTCH1 ID protein levels (Appendix A). KRAS silencing was also accompanied by a significant downregulation of ABCG2 and HES1, target genes of HH-GLI and NOTCH1 ID, respectively (Appendix A). 

Arsenic Trioxide (ATO) is an organic compound approved for the therapy of adult patients with acute promyelocytic leukemia [25] and was shown to successfully inhibit both HH-GLI and NOTCH pathways [26]. ATO’sability to inhibit both GLI1 and NOTCH ID levels was confirmed in the KRASG13D-driven CRC model (Figure 1D). As previously shown, 5-FU alone was able to upregulate both GLI1 and NOTCH1 ID, while the combination with ATO impaired both signaling pathways (Figure 1D). Cleaved-PARP levels showed that apoptosis was significantly increased by the combination of ATO and 5-FU, while we observed a non-significant trend in ATO-treated cells (Figure 1D).

A pivotal feature of CRC aggressiveness relies on the epithelial-to-mesenchymal transition (EMT), a process that includes the acquisition by cancer cells of properties including motility and migration, early steps in cancer invasion and metastasis.

Therefore, we investigated whether the targeting of HH-GLI and NOTCH could impair KRAS mutant CRC’s migratory ability.

We investigated the effects of the combined treatment of 5-FU and ATO on the migration ability of HCT116 cells. We observed that the migration was unaffected by 5-FU treatment, while it was impaired with ATO treatment and was completely abrogated after ATO plus 5-FU combined treatment (Figure 1E). Then, we evaluated epithelial differentiation through E-cadherin levels, which increased after the combined treatment of ATO and 5-FU (Appendix A).

### 3.2. HH-GLI Signaling Pathway Sustains Resistance to 5-FU in BRAF Mutant CRC Cells

BRAF V600E is the activating driving mutation in 10% of CRC and correlates with poor prognosis, however targeted therapy against the mutation was proven ineffective and first-line treatment includes cytotoxic chemotherapy [1]; thus, we investigated the role of the HH-GLI and NOTCH signaling pathways in 5-FU chemotherapy resistance in BRAF mutant HT29 cells.

HT29 cells were treated with 5-FU alone or in combination with the HH-GLI inhibitor GANT61 and the NOTCH1 inhibitor DAPT (Figure 2A). We observed that 5-FU induced upregulation of GLI1 and NOTCH1. GANT61 treatment resulted in the downregulation of both GLI1 and NOTCH1 ID, while DAPT treatment caused the downregulation only of NOTCH1 ID, without exerting any effect on GLI1 levels compared to control cells. The combination of GANT61 and DAPT successfully targeted both GLI1 and NOTCH1 ID. The combined treatment of GANT61 plus 5-FU was able to revert the 5-FU-induced upregulation of GLI1 and NOTCH1 ID, and the combined treatment of DAPT and 5-FU was able to revert the 5-FU-induced upregulation of NOTCH1 ID and partially of GLI1. Only when both HH-GLI and NOTCH pathways were inhibited together with 5-FU treatment were both GLI1 and NOTCH1 ID significantly downregulated (Figure 2A).

Apoptosis was evaluated through c-PARP levels; treatment with 5-FU and single inhibition of HH-GLI and NOTCH1 failed to induce apoptosis; c-PARP levels indeed increased only when cells were treated with GANT61 in combination with 5-FU, or with the combination of the three drugs (Figure 2A).

We then investigated cell viability and our results showed a significant impairment after GANT61 treatment, alone or in combination with 5-FU (Figure 2B).

Based on these results, chemotherapy resistance to apoptosis in BRAF V600E mutated cells seems to be driven by the HH-GLI signaling, which in turn sustains the activation of the NOTCH pathway. To gain more insight into the interdependence between the HH-GLI and NOTCH pathways, we interrogated GLI1 and NOTCH1 levels in a cohort of CRC patients carrying *BRAFV600E* mutation (Mutation status (Core Exon)—Sieber—211—rma_sketch—huex10p; https://hgserver1.amc.nl/cgi-bin/r2/main.cgi, accessed on 15 December 2021) and found a positive and significant correlation between GLI1 and NOTCH1 (Figure 2C). The above presented data suggest an upstream role of HH-GLI in the regulation of NOTCH signaling in the BRAF-driven CRC model. 

To investigate whether BRAFV600E acted as a driver on the regulation of HH-GLI and NOTCH, we performed BRAF silencing (Appendix A). BRAF silencing resulted in decreased GLI1 and NOTCH1 ID protein levels (Appendix A). 

We also evaluated mRNA levels of HH-GLI and NOTCH1 ID readout, ABCG2 and HES1, respectively, and both were significantly decreased after BRAF silencing (Appendix A).

The above-reported data demonstrate that chemotherapy stress induced increased levels of both HH-GLI and NOTCH1 pathways in the BRAF-driven CRC model. Interestingly, we observed that the GLI1 inhibitor GANT61 was also able to decrease NOTCH1 ID levels; conversely, the NOTCH1 inhibitor DAPT did not affect GLI1 levels.

Since we observed that the targeting of HH-GLI was able to indirectly also target the NOTCH pathway, we wondered if the combination of 5-FU and GANT61 could affect cell motility, a key feature of EMT and therefore of CRC aggressiveness. 

Our experiments showed that 5-FU did not affect cell motility, while GANT61 resulted in decreased cell motility, which was further impaired by the combination of GANT61 with 5-FU (Figure 2D). 

Then, we investigated the expression of two HT29 cell-specific epithelial differentiation markers, Axin and Muc2. We observed upregulation of Axin only after combined treatment, while Muc2 was affected by both 5-FU and GANT61 alone and by their combination (Appendix A), suggesting that treatments enhance the differentiated phenotype. 

### 3.3. 5-FU Increases Motility of CRC Organoids

The previous set of experiments allowed us to point out the role of HH-GLI and NOTCH pathways as regulators of EMT in KRAS mutant and BRAF mutant CRC, a key feature of chemoresistance [27]. Organoid models in pre-clinical studies have become widespread due to their high reproducibility and high similarity to in vivo models [28,29]. Indeed, cell features and behavior depend on the architecture of the cell population, e.g., the cell–cell contact, the stiffness of the extracellular matrix and the interaction with the microenvironment. All these conditions concur with specific characteristics related to cell polarity, stemness and differentiation status.

Thus, to obtain CRC organoids, we seeded HCT116 and HT29 cells in Matrigel and after 7 days we observed organoid growth, as shown in Figure 3A,B. We compared basal levels of GLI1 and NOTCH1 in organoids and in 2D monolayer and our results reported higher GLI1 and HES1 expression levels in organoids, indicating that both pathways were more active in organoids compared to monolayer cellular models (Figure 3C). We then evaluated levels of the EMT marker c-MET in both organoids and monolayers and observed that c-MET was expressed at higher levels in organoids (Figure 3C). Since our results showed that 5-FU was not able to impair the migratory ability of CRC (Figure 1 and Figure 2), and that CRC patients often present disease progression despite chemotherapy, we wondered if 5-FU itself favored aggressiveness in organoids, unleashing the migratory potential.

Increased motility and migration capacity are features of EMT, thus we performed in vivo live cell imaging in the KRASG13D-driven CRC organoid model, the HCT116-derived organoids (oHCT116) at basal state and after 5-FU treatment (Appendix A).

To investigate the behavior of CRC cells within organoids, we investigated the diffusion parameters that allow the motility of individual cells to be quantified. The diffusion parameters from the oHCT116 control or 5-FU-treated organoids are reported (Figure 3E) along with the single cell trajectories that were used for the calculation of the diffusion parameters (Appendix A). Interestingly, 5-FU-treated oHCT116 cells mostly present lower diffusion parameters compared with CTRL (Figure 3D), with a long tail corresponding to a sub-group of cells presenting very high diffusion (Figure 3E). Based on these results, we believe that cells with augmented motility after chemotherapy represent a subset of aggressive cells able to initiate the metastatic process.

### 3.4. HH-GLI and NOTCH Inhibition Impairs 5-FU-Driven Mesenchymal Phenotype in KRASG13D-Driven CRC Organoids

We then proceeded to investigate the inhibition of HH-GLI and NOTCH by using ATO in combination with 5-FU in KRAS-driven CRC organoids, oHCT116.

Treatment with 5-FU alone did not affect organoid growth, while organoids treated with ATO were significantly smaller; the association of 5-FU and ATO further impaired organoid growth (Figure 4A). Expression levels of the EMT marker *c-MET*, cancer stemness markers *ABCG2* and *CD133a,* which is both HH-GLI target and cancer stemness marker, were significantly decreased in the combined treatment of 5-FU and ATO (Figure 4B). Interestingly, ATO was able to counteract the 5-FU-driven upregulation of ABCG2.

Mesenchymal features were also investigated by the immunofluorescence of the EMT marker vimentin, whose levels increased after 5-FU treatment and were reduced when organoids were treated with ATO alone or in combination with 5-FU. Of note F-actin, revealed by phalloidin staining, underwent a marked rearrangement in 5-FU-treated oHCT116, where cells lost their pseudopodia, probably due to a modification in the cell polarity (Figure 4C).

### 3.5. HH-GLI Inhibition Impairs 5-FU-Driven Mesenchymal Phenotype in BRAFV600E-Driven CRC Organoids

We then investigated the effects of 5-FU alone or in combination with the HH-GLI blockade in the BRAFV600E-driven CRC organoids (oHT29).

Our results showed that the size of oHT29 treated with 5-FU did not differ from the control group, while organoids treated with GANT61 were smaller in size and the combination of 5-FU and GANT61 strongly impaired organoid growth (Figure 5A).

Gene expression analysis showed that the levels of cancer stem cell and EMT markers *ABCG2, CD133* and *c-MET* significantly increased after chemotherapy treatment and were impaired by HH-GLI inhibition and the combination of 5-FU and GANT61 (Figure 5B).

To better investigate EMT, we performed whole-mount immunofluorescence staining for the mesenchymal marker vimentin and observed that vimentin levels were upregulated in 5-FU-treated organoids, they decreased with GANT61 and were strongly impaired in the combined treatment (Figure 5C).

Altogether, our experiments show that in KRAS-driven and BRAF-driven CRC, the HH-GLI and NOTCH pathways sustain the resistance to 5-FU through the activation of the EMT. Of note, ATO, the drug targeting both HH-GLI and NOTCH pathways, reverted the mesenchymal phenotype, therefore supporting the action of the chemotherapeutic drug.

## 4. Discussion

Despite recent advances in cancer therapy, CRC is still among the prevalent causes of cancer-related death [30]. Even though medical research has focused on identifying genetic mutations linked to CRC progression and tumor prognosis to improve patient treatment, drug resistance often occurs. One of the mechanisms conferring drug resistance is the misactivation of evolutionarily conserved pathways, such as Wingless (WNT) [31,32], phosphoinositide-3-kinase [33,34], extracellular signal-regulated kinase (ERK) [35,36], nuclear factor-κB (NF-κB) [37,38] and the Hedgehog-GLI (HH-GLI) signaling pathway [20]. The HH-GLI pathway has a crucial role in correct embryonic development and plays a role in the physiological maintenance of many tissues, including the colonic mucosa [39,40]. While canonical activation of the HH-GLI pathway transduces the signal through the Hedgehog/PTCH/SMO/GLI axis, non-canonical regulation of GLI is external to Hedgehog signaling. Of note, it was demonstrated that transforming growth factor-beta (TGF-β) [41], epidermal growth factor receptor (EGFR) [42], mitogen-activated protein kinases (MAPK) [11], β-arrestin [43] and WNT/β-catenin [44,45] were able to induce the expression of GLI, regardless of SMO activation. Since both canonical and non-canonical routes culminate with the activation of the GLI1 transcriptional program, GLI1 inhibition could be useful to prevent chemoresistance in cancer cells. Our group has previously demonstrated that HH-GLI signaling regulates the expression of ATP-binding cassette transporters (ABC transporters), which are correlated to multidrug resistance in cancer cells, providing a rationale for the consideration of the HH-GLI pathway as a therapeutic target in CRC [20]. NOTCH signaling has been reported to play a crucial role in the development of the normal mucosa [15] and its aberrant activation is related to carcinogenesis in CRC. HH-GLI and NOTCH signaling pathways together with the WNT and BMP pathways are responsible for the development of intestinal mucosa, which is the innermost layer of the colon. Stem cells, transit amplifying cells and terminally differentiated secretory cells or enterocytes, concur in the formation of the structural unit of the colon, known as the crypt of Lieberkuhn [46]. A recent paper showed that the HH-GLI blockade with GANT61 was able to inhibit NOTCH and WNT/β-catenin in cellular models of CRC [47]. Since the HH-GLI and NOTCH pathways play a fundamental role in the correct patterning of the colonic mucosa and HH-GLI is upregulated by chemotherapeutic stress, we wondered whether HH-GLI and NOTCH crosstalk could be involved in the resistance mechanism of CRC cells related to 5-FU chemotherapeutic stress.

The results of this study show how the HH-GLI and NOTCH pathways sustain CRC chemoresistance in different ways depending on the driver oncogene mutation. In detail, in KRASG13D-driven HCT116 cells we observed an upregulation of HH-GLI and NOTCH pathways after 5-FU and the inhibition of HH-GLI resulted in increased levels of NOTCH1 ID and vice versa (Figure 1A). These results, coupled with the interrogation of public datasets (Figure 1C) suggested that the HH-GLI and NOTCH signaling pathways are connected in a positive feedback loop aiming to escape apoptosis induced by 5-FU (Figure 6).

Importantly, the combined inhibition of HH-GLI and NOTCH was able to impair EMT, shown both as an impairment of transwell migration ability and with EMT markers in organoids (Figure 1E and Figure 4). ATO, which was used to target both HH-GLI and NOTCH pathways, has been approved by the FDA for the therapy of adult patients with acute promyelocytic leukemia (APL). A phase I trial investigating the co-administration of ATO and 5-FU/Leucovorin in patients with advanced/relapsed CRC showed that ATO was well tolerated and that in some patients it was associated with therapeutic response and increased survival; a later study investigated GLI1 levels in biopsies from the above-mentioned clinical trial and found that it resulted to be down-modulated after ATO administration. Of note, data on the mutational status of enrolled patients are not available [48,49]. In BRAFV600E-driven CRC, both pathways were upregulated after 5-FU treatment, but importantly GANT61 downregulated not only its specific target GLI1 but also NOTCH (Figure 2A), suggesting an upstream role of HH-GLI over the NOTCH pathway (Figure 6), thus explaining the positive correlation between these two signaling pathways (Figure 2C). Importantly, HH-GLI inhibition was able to impair EMT features, both in monolayer and organoids (Figure 2D and Figure 5).

## 5. Conclusions

In conclusion, our study describes for the first time two distinct models for KRAS- and BRAF-driven CRC where the HH-GLI and NOTCH signaling pathways play different roles in the chemoresistance and mesenchymal phenotype of CRC (Figure 6). Indeed, we described that in KRASG13D-driven CRC, chemotherapy resistance is directed by the concurrent activation of the HH-GLI and NOTCH pathways and the inhibition of both is crucial to revert the resistant phenotype. Conversely, in BRAFV600E-mutated CRC, the resistance to apoptosis induced by chemotherapy is mainly sustained by the HH-GLI signaling pathway. The implications of this novel information can be far-reaching if taken into consideration for the management of CRC patients, providing clinicians with further tools for the development of more effective treatment plans.

## Figures and Tables

**Figure 1 cancers-15-01471-f001:**
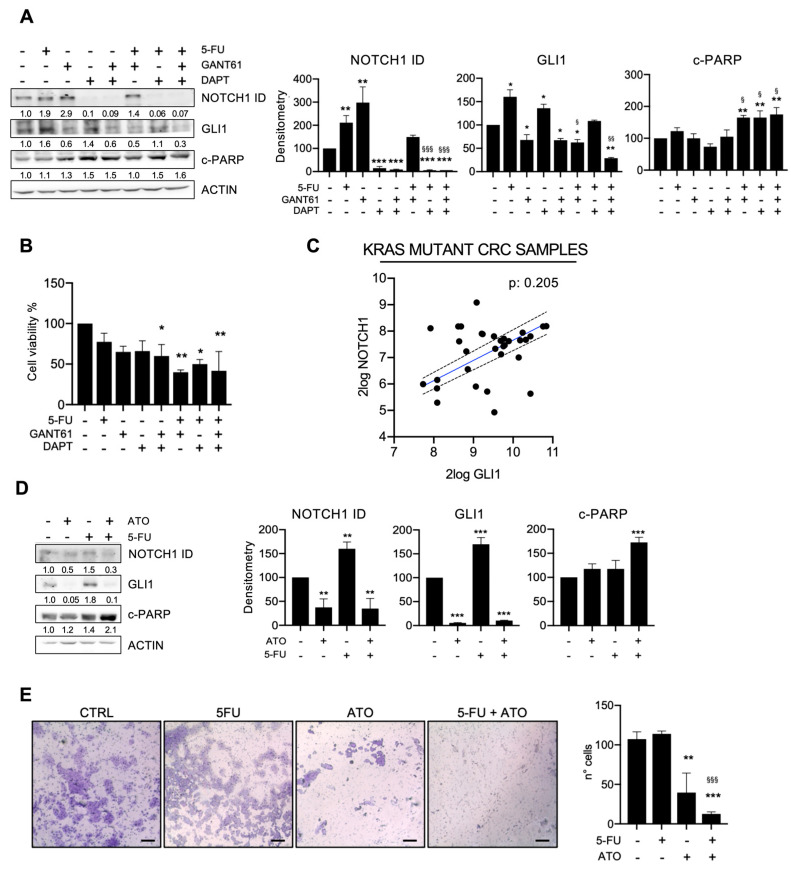
HH-GLI and NOTCH signaling pathways sustain resistance to 5-FU in KRAS mutant CRC cells. (**A**) Western blot analysis of NOTCH1 Intracellular domain (NOTCH1 ID), GLI1 and cleaved-PARP (c-PARP) in HCT116 cells after 5-FU treatment in combination with GANT61 and DAPT. Numbers indicate intensity ratio of bands. Bar graphs show densitometric quantification of the band intensity values normalized to the loading control. * *p*  <  0.05; ** *p*  <  0.01; *** *p* < 0.001 versus control; ^§^
*p* < 0.05 versus 5-FU; ^§§^
*p*  <  0.01; ^§§§^
*p* < 0.001 (Two-way ANOVA test). Uncropped full scan in Appendix A. (**B**) Evaluation of cell viability by Trypan Blue exclusion assay in HCT116 after 5-FU treatment with or without GANT61 and/or DAPT; * *p*  <  0.05, ** *p*  <  0.01 versus CTRL (One-way ANOVA test)**.** (**C**) Correlation analysis between GLI1 and NOTCH1 expression from dataset interrogated on R2 platform, as indicated in main text. (**D**) Western blot analysis of GLI1, NOTCH1 ID and c-PARP in HCT116 cells treated with 5-FU and ATO. Numbers indicate intensity ratio of bands. Bar graphs show densitometric quantification of the band intensity values normalized to the loading control; * *p*  <  0.05; ** *p*  <  0.01; (Two-way ANOVA test). (**E**) Transwell invasion assay in HCT116 cells treated with 5-FU, ATO and the combined treatment and control group (CTRL). Scale bar 150 µm. ** *p*  < 0.01; *** *p* < 0.001; versus control; ^§§§^
*p* < 0.001 (One-way ANOVA test).

**Figure 2 cancers-15-01471-f002:**
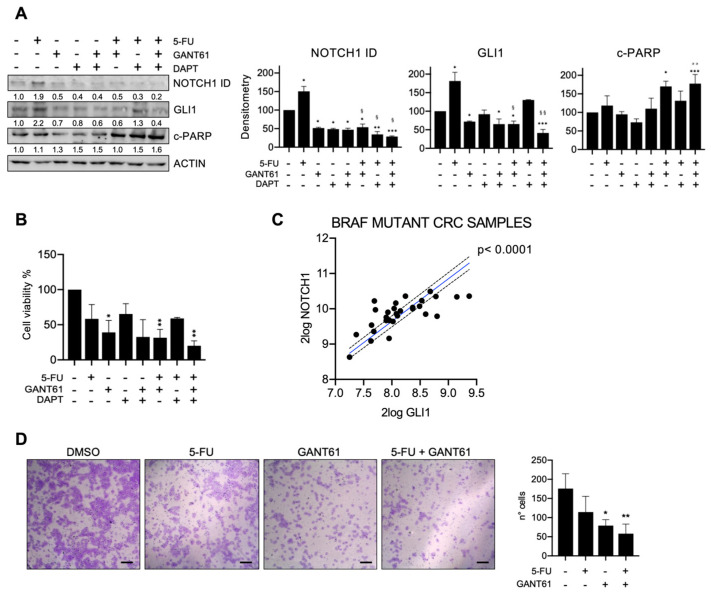
HH-GLI signaling pathway sustains resistance to 5-FU in BRAF mutant CRC cells. (**A**) Western blot analysis of NOTCH1 Intracellular domain (NOTCH1 ID), GLI1 and cleaved-PARP (c-PARP) in HT29 cells after 5-FU treatment in combination with GANT61 and DAPT. Numbers indicate intensity ratio of bands. Bar graphs show densitometrically quantified band intensity values normalized to the loading control; * *p*  <  0.05 ** *p*  <  0.01; *** *p* < 0.001; ^§^
*p* < 0.05 versus 5-FU; ^§§^
*p* < 0.01 (Two-way ANOVA test). Uncropped full scan in Appendix A. (**B**) Evaluation of cell viability by Tripan Blue exclusion assay in HT29 after 5-FU treatment in combination with GANT61 and DAPT; * *p*  <  0.05, ** *p*  <  0.01 versus CTRL (One-way ANOVA test). (**C**) Correlation analysis between GLI1 and NOTCH1 expression from dataset interrogated on R2 platform as indicated in main text. (**D**) Transwell invasion assay in HT29 cells treated with 5-FU, GANT61, the combined treatment and control group (CTRL); Scale bar 150 µm; * *p*  <  0.05, ** *p*  <  0.01 versus CTRL (One-way ANOVA test).

**Figure 3 cancers-15-01471-f003:**
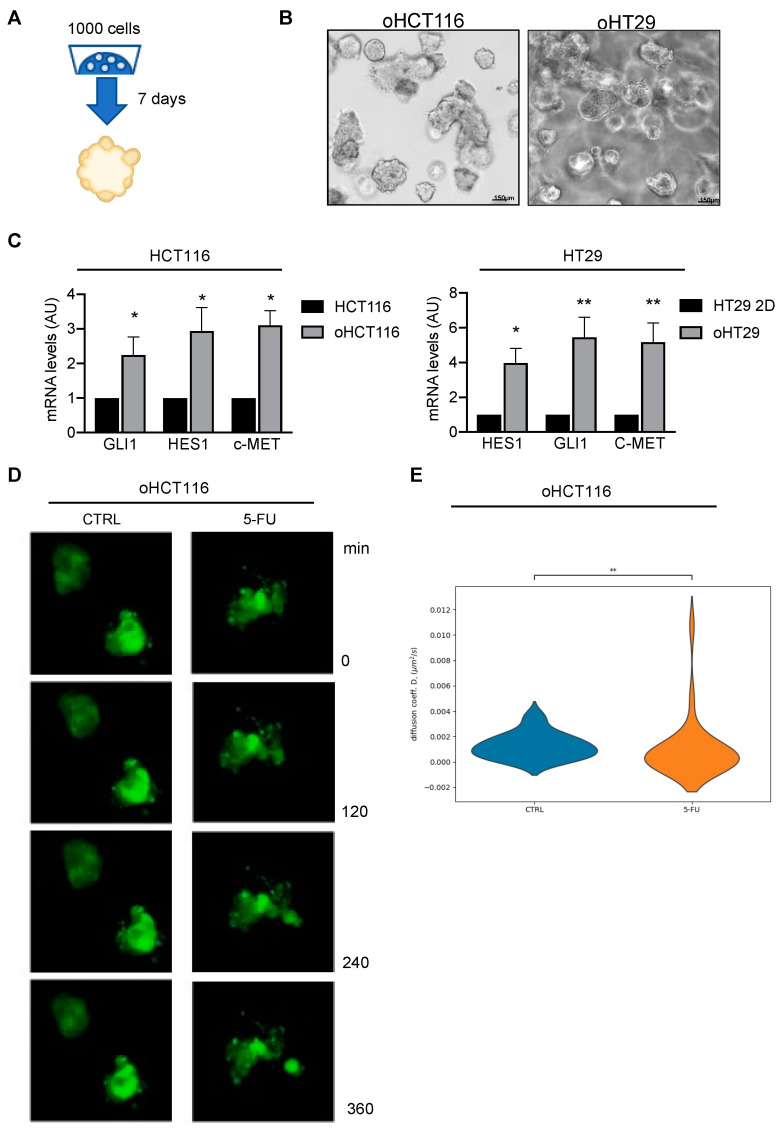
5-FU increases motility in CRC organoids. (**A**) Workflow for CRC organoid growth. (**B**) Brightfield image of HCT116 and HT29 organoids (respectively oHCT116 and oHT29) after 7 days of culture. (**C**) Quantitative real-time PCR of HES1, GLI1, c-MET expressed in HCT116 and HT29 cultured in monolayer and as organoids; * *p*  <  0.05 versus CTRL; ** *p*  <  0.01 (Two-way ANOVA test). (**D**) Fluorescent images of GFP-labeled HCT116 on sequential hours, scale bar 200 µm. Appendix A of time lapse experiments are available in Appendix A. (**E**) Violin plot of the diffusion parameters obtained from the single cell trajectories for CTRL and 5-FU-treated organoids. Kolmogorov–Smirnov test *p*-value 0.0032.

**Figure 4 cancers-15-01471-f004:**
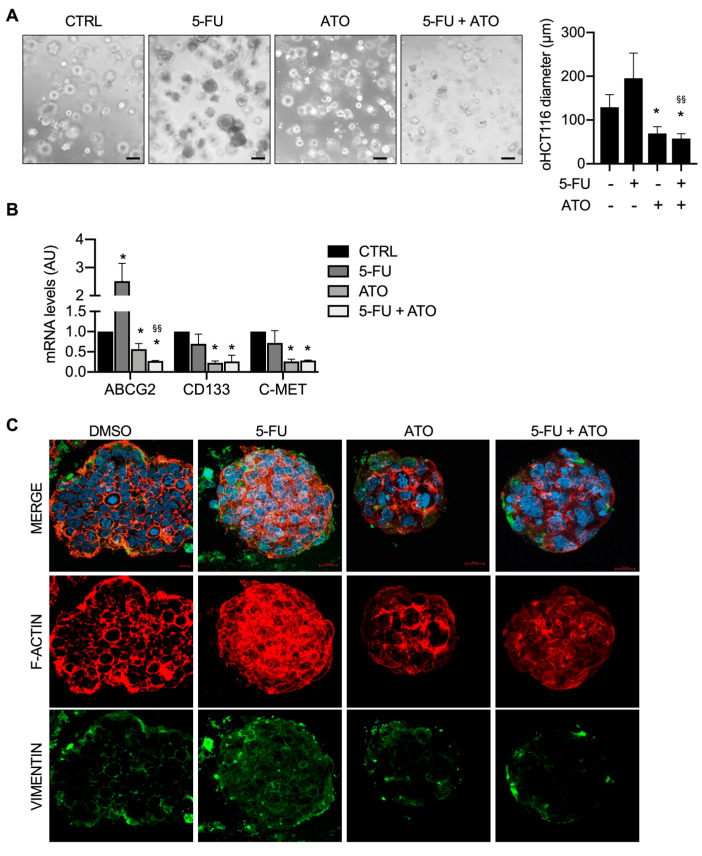
HH-GLI and NOTCH inhibition impairs 5-FU-driven mesenchymal phenotype in *KRASG13D*-driven CRC organoids. (**A**) Brightfield image of oHCT116 treated with 5-FU, ATO, their combination and the control group (CTRL); Scale bar 150 µm; * *p*  <  0.05 versus CTRL; ^§§^
*p* < 0.01 versus 5-FU (One-way ANOVA test). (**B**) Quantitative real-time PCR of stem markers expressed in oHCT116 treated with 5-FU, ATO, their combination and the control group (CTRL). mRNA levels of ABCG2, CD133a and c-MET expressed in oHCT116 were expressed in arbitrary units; * *p*  <  0.05 versus CTRL; ^§§^
*p* < 0.01 versus 5-FU (Two-way ANOVA test). (**C**) Whole-mount immunofluorescence staining of oHCT116 stained with phalloidin-594 (F-actin, red), vimentin (mesenchymal marker, green) and DNA (DAPI). Images were analyzed by using the program Zeiss ZEN 2.3 blue edition. Scale bar 200 µm.

**Figure 5 cancers-15-01471-f005:**
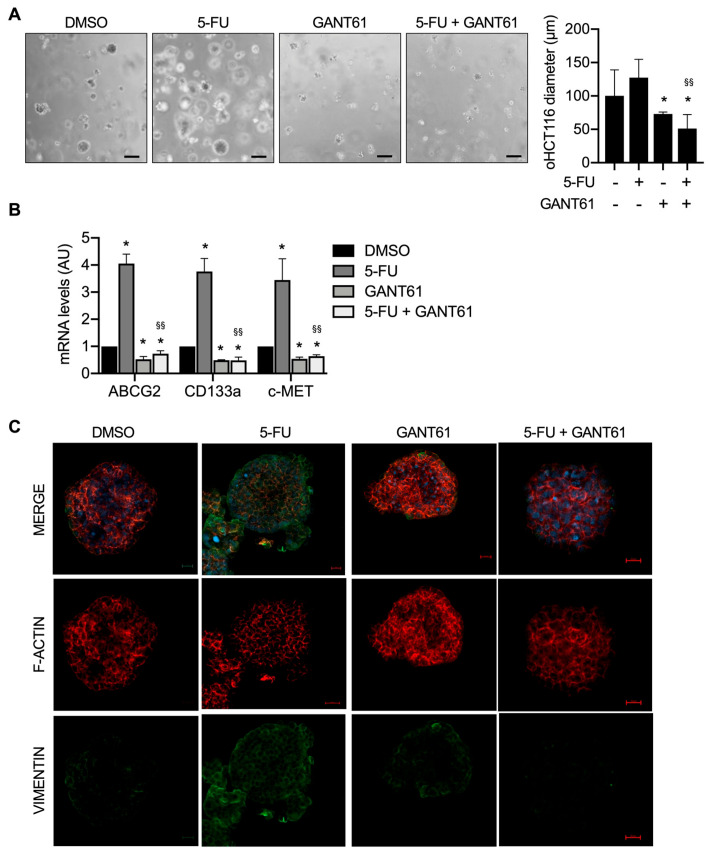
HH-GLI inhibition impairs 5-FU-driven mesenchymal phenotype in *BRAFV600E*-driven CRC organoids. (**A**) Brightfield image of oHT29 treated with 5-FU, GANT61, their combination and the control group (CTRL); Scale bar 150µm; * *p*  <  0.05 versus Ctrl; ^§§^
*p* < 0.01 versus 5-FU (One-way ANOVA test). (**B**) Quantitative real-time PCR of ABCG2, CD133a and c-MET expressed in oHT29 were expressed in arbitrary units. Data are representative of three independent experiments, * *p*  <  0.05 versus Ctrl; ^§§^
*p* < 0.01 versus 5-FU (Two-way ANOVA test). (**C**) Whole-mount immunofluorescence of HT29 organoids stained using phalloidin-594 (f-actin, red), vimentin (mesenchymal marker, green) and DNA (DAPI). Images were analyzed by using the program Zeiss ZEN 2.3 blue edition. Scale bar 200 µm.

**Figure 6 cancers-15-01471-f006:**
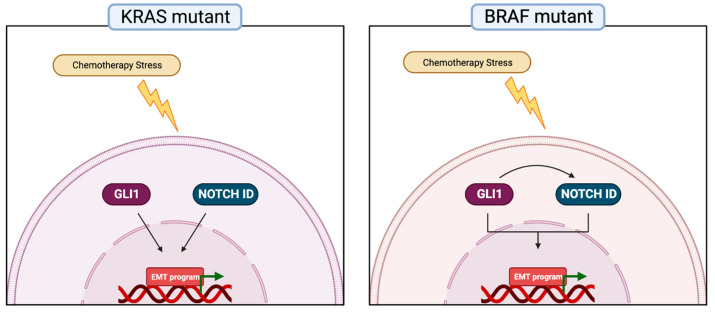
Hedgehog-Gli and NOTCH pathways sustain chemoresistance and the mesenchymal phenotype in CRC. Model of the activity of the Hedgehog-GLI and NOTCH pathways after chemotherapy stress in BRAFV600E and KRASG13D models. In KRAS-driven CRC, the chemotherapy stress activates both HH-GLI and NOTCH, which independently sustain the EMT program; in BRAF-driven CRC, chemotherapy stress induces the activation of the HH-GLI pathway, which in turn sustains the activation of NOTCH1 signaling, determining the acquisition of the EMT phenotype.

## Data Availability

All relevant data are included in the manuscript.

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
