# Peer review of "Hedgehog-GLI and Notch Pathways Sustain Chemoresistance and Invasiveness in Colorectal Cancer and Their Inhibition Restores Chemotherapy Efficacy"

_cancers, 2023, doi:10.3390/cancers15051471_

Round 1

Reviewer 1 Report

The manuscript entitled “Hedgehog-GLI and Notch pathways sustain chemoresistance and invasiveness in colorectal cancer and their inhibition restores chemotherapy efficacy” is a well-written and well-designed study. This work evaluated the role of HH-GLI and NOTCH signaling pathways as molecular mechanisms responsible for chemotherapy resistance to 5-FU. However, a few points need to be addressed:

-       In the Material and Methods section, the sentence “Cell lines were obtained from the American Type Culture Collection (ATCC).” (page 2, lines 91-92) should be removed. The origin of cell lines is already described in lines 88 to 89.

-       The Authors used trypan blue assay to evaluate cell proliferation. However, how was cell proliferation calculated? Cell proliferation is usually presented as cell density (cell/volume) and cell viability as a percentage, but the Authors showed cell proliferation as a percentage (Figures 1B and 2B).

-       On page 3 (line 132), “in vivo” should be italicized.

-       In Figure 4A, GANT61 should be replaced by ATO in the oHCT116 diameter graphic.

-       In Figure 6, GLI and NOTCH should be in uppercase.

Author Response

The manuscript entitled “Hedgehog-GLI and Notch pathways sustain chemoresistance and invasiveness in colorectal cancer and their inhibition restores chemotherapy efficacy” is a well-written and well-designed study. This work evaluated the role of HH-GLI and NOTCH signaling pathways as molecular mechanisms responsible for chemotherapy resistance to 5-FU.

Authors’ reply: We thank the reviewer for appreciating our work and we revised the manuscript according to their suggestions.

However, a few points need to be addressed:

-       In the Material and Methods section, the sentence “Cell lines were obtained from the American Type Culture Collection (ATCC).” (page 2, lines 91-92) should be removed. The origin of cell lines is already described in lines 88 to 89.

Authors’ reply: We fixed the mistake.

-       The Authors used trypan blue assay to evaluate cell proliferation. However, how was cell proliferation calculated? Cell proliferation is usually presented as cell density (cell/volume) and cell viability as a percentage, but the Authors showed cell proliferation as a percentage (Figures 1B and 2B).

Authors’ reply: we thank the reviewer for highlighting this issue. After discussing within the authors group, we replaced “proliferation” with “viability” in the methods and in the text and figures where appropriate. In addition, we included an explanation on how the viability was calculated. (lines 113 – 117)

-       On page 3 (line 132), “in vivo” should be italicized.

Authors’ reply: We fixed the mistake.

-       In Figure 4A, GANT61 should be replaced by ATO in the oHCT116 diameter graphic.

Authors’ reply: We fixed the mistake.

-       In Figure 6, GLI and NOTCH should be in uppercase.

Authors’ reply: We fixed the mistake.

Reviewer 2 Report

Authors of the work "Hedgehog-GLI and Notch pathways sustain chemoresistance and invasiveness in colorectal cancer and their inhibition restores chemotherapy efficacy" presented very intersting work not only from the basic science point of view but also from clinical perspective.

Hovewer The introdusction must be improved and extended.

Moreover if the non-parametric tests are used so the results should be expressed ad median not mean and interquartile range or min/max instead of SD.

Additionally refferences are quite old, and not cited as journal require so improve the refferences ans up to date them.

Author Response

Authors of the work "Hedgehog-GLI and Notch pathways sustain chemoresistance and invasiveness in colorectal cancer and their inhibition restores chemotherapy efficacy" presented very intersting work not only from the basic science point of view but also from clinical perspective.

Authors’ reply: We thank the reviewer for appreciating our work and we revised the manuscript according to their suggestions.

Hovewer The introdusction must be improved and extended.

Authors’ reply: We thank the reviewer for raising this point. Introduction was enriched with additional information, including additional references, to better clarify the state of the art that led to our experimental plan. New parts in the revised manuscript are: lines 57 – 59; 65 – 68; 82 – 86; 89 – 92.

Moreover if the non-parametric tests are used so the results should be expressed ad median not mean and interquartile range or min/max instead of SD.

Authors’ reply: we apologize for including not relevant parts in the method section and we thank the reviewer for pointing that out. After carefully re-checking of methods applied to all experiments, we removed the sentence relative to non-parametric tests. This can be found in the revised manuscript, lines 189 – 192. We included the indication of the statistical test we used in every figure legend. 

Additionally refferences are quite old, and not cited as journal require so improve the refferences ans up to date them.

Authors’ reply: we thank the reviewer for this observation, that prompted us to improve the introduction and discussion sections. New refs were added in the revised manuscript, and we updated the format.

new references in the revised manuscript: 7, 8, 9, 13, 14, 16, 17, 21, 43, 47.

Reviewer 3 Report

In this manuscript, the authors demonstrated that two distinct models for KRAS- and BRAF-driven CRC where HH-GLI and NOTCH signaling pathways play different roles in the chemoresistance and mesenchymal phenotype of CRC for the first time. The results providing clinicians with further tools for the development of more effective treatment plans. This is an interesting study and the data presented here are solid. Therefore, this manuscript seems acceptable in the journal of Cancers. But there are some comments that should be clarified.

No.1: The relationship between HH-GLI and NOTCH signaling pathway should be described more clearly.

No.2: The result part should be simplified and more logical.

Author Response

In this manuscript, the authors demonstrated that two distinct models for KRAS- and BRAF-driven CRC where HH-GLI and NOTCH signaling pathways play different roles in the chemoresistance and mesenchymal phenotype of CRC for the first time. The results providing clinicians with further tools for the development of more effective treatment plans. This is an interesting study and the data presented here are solid. Therefore, this manuscript seems acceptable in the journal of “Cancers”. But there are some comments that should be clarified.

Authors’ reply: We thank the reviewer for appreciating our work and we revised the manuscript according to their suggestions.

No.1: The relationship between HH-GLI and NOTCH signaling pathway should be described more clearly.

Authors’reply: We thank the reviewer for raising this point. We included new description and refs about this topic in introduction and discussion. Lines: 89 – 92; lines: 482 – 483. New references: 16, 17, 47

No.2: The result part should be simplified and more logical.

Authors’reply: we thank the reviewer for the comment, and we revised the result section to improve clarity and ease of understanding.
